# Robustness of explainable AI algorithms for disease biomarker discovery from functional connectivity datasets

Deepank Girish[1]*, Yi Hao Chan[1]*, Sukrit Gupta[2], Jing Xia[1] and Jagath C. Rajapakse[1]

*Abstract*—Machine learning (ML) models have been used in functional neuroimaging for wide-ranging tasks, ranging from disease diagnosis to disease prognosis. There have been successive functional connectivity-based ML studies focused on improving model performances for disease detection. An increasing number of such studies use their trained models to detect and evaluate salient features that could be potential biomarkers of these neurological conditions. The evaluation of these salient features is often qualitative and limited to cross-referencing existing literature for similar findings. In this study, we present objective quantitative metrics to evaluate the robustness of these salient features. Building upon existing generic evaluation metrics, we propose metrics that capture topological properties known to be characteristic of brain functional connectomes. Using existing and newly proposed measures on a set of baselines and state-of-the-art graph neural networks (GNN) models, we found that when GNNExplainer is used with models that incorporate attention, the scores produced are relatively more robust than other combinations. On datasets of patients with Autism Spectrum Disorder (ASD) or Attention-deficit Hyperactivity Disorder (ADHD), our proposed metrics highlighted that salient features identified in both disorders are highly involved in functional specialization, while salient ASD features expressed stronger functional integration than ADHD. We package these existing and novel metrics together in the RE-CONFIRM framework that holds promise to set the foundations for the quantitative evaluation of salient features detected by future studies.

*Index Terms*—Biomarker Discovery, Graph Neural Networks, Functional Connectivity, Model Explainability

## I. INTRODUCTION

Since its inception almost three decades ago, functional magnetic resonance imaging (fMRI) has been widely used to further our understanding of varied brain states, including multiple neurological conditions. Neurological conditions often have poorly understood etiologies, but the presence of modulations in functional connectivity (FC) are well documented [1]. Functional MRI produces 4-dimensional information and often comes along with rich metadata (e.g. clinical test scores) making manual analysis infeasible. This motivates the use of Machine Learning (ML) approaches to model fMRI datasets, which has since evolved into the use of deep learning architectures such as GNN [2] and transformers [3]. In these studies, the predominant approach is to train models for classifying

between healthy subjects and patients. These models have been criticized for their poor generalization abilities and their black-box nature [4]. Thus, the utility of such classifiers in clinical settings remains very limited. To ameliorate these issues, researchers have used model interpretability techniques to give clinicians insights on the model's decisions and biomarker discovery [5], [6].

Given a GNN model with acceptable classification performance, interpretability methods called 'explainers', ranging from post-hoc techniques such as Integrated Gradients [7] and GNNExplainer [8] to intrinsically interpretable models such as the attention mechanism in Graph Attention Networks (GAT) [9] and pooling layers in BrainGNN [10] can be applied on the model. Thereafter, the potential biomarkers uncovered by the explainer are evaluated for their correctness. These evaluations are currently mostly limited to arbitrarily chosen features (e.g. top-$K$ features with the highest importance scores) with cross-references being made to existing literature. Such evaluations do not account for the possibility that the features highlighted by both studies could be erroneous. Therefore, better metrics are needed to quantify the robustness of these salient features.

A recent survey paper by Nauta et al. [11] has consolidated 12 key properties of explainers which they name 'Co-12' (e.g. correctness, consistency, contrastivity, etc.). However, not all existing metrics are applicable in the context of disease biomarker discovery. For instance, ground truth disease biomarkers are often not available, making it challenging to determine the *coherence* of the explanation with existing knowledge. Further work is needed to determine which metrics are relevant for biomarker discovery. Graph-based evaluation metrics have also been proposed [12], [13], including metrics such as graph explanation stability, which checks for consistency of explanations when the input graph is perturbed. However, they do not make use of known properties of connectomes such as modularity of FC and the presence of functional hubs. Finally, while meta-analyses can be done to consolidate reported biomarkers and identify robust biomarkers, our proposed approach provides quantitative measures that can improve future meta-analyses on disease biomarkers.

In this study, we identify evaluation metrics relevant to FC and also propose novel evaluation metrics to objectively measure the robustness of saliency scores produced by existing explainers in the context of discovering FC biomarkers. Modularity is a well-established concept in FC studies and is understood to be affected by diseases [14]. We propose

[1]Deepank Girish, Yi Hao Chan, Jing Xia and Jagath C. Rajapakse are with the College of Computing and Data Science, Nanyang Technological University, Singapore.
[2]Sukrit Gupta is with the Department of Computer Science and Engineering, Indian Institute of Technology Ropar, India.
* These authors contributed equally.

modular ratio, a metric that measures the extent of similarity between the saliency scores of nodes in the same module, relative to other nodes in the functional brain network. Furthermore, several neurological disorders such as ASD and ADHD are known to affect hubs [15], Thus, we propose a measure of assortativity that accounts for hubs, quantifying the extent to which hubs are captured by the saliency scores.

Putting these novel metrics together with our selection of existing metrics that are most relevant to FC studies, we propose Robust Evaluation of CONnectome Features Identified by Relevance Measures (RE-CONFIRM), a framework that could be used by future disease classification studies to further evaluate their salient features in a more robust and objective manner. While existing frameworks like Quantus [16] and OpenXAI [17] offer implementations of evaluation metrics and benchmarks for assessing explanation methods on tabular and image datasets, they do not address the evaluation of explainability in the context of brain connectomes using fMRI data, nor do they interpret scores in terms of biomarkers.

Applying RE-CONFIRM to the ABIDE and ADHD-200 datasets, we found that models incorporating attention tend to be more robust than models that do not, especially when these models are used with GNNExplainer. Additionally, salient features of ASD and ADHD have low modular ratios, suggesting that they exhibit strong modular relationships. Hub assortativity scores further reveal stronger functional integration in ASD than in ADHD. The key contributions of this study are:

- A framework to evaluate the robustness of salient FC features detected by different explainers. Future studies can use RE-CONFIRM to evaluate the potential biomarkers they discovered in a more robust manner.
- Focusing on the brain's functional connectome, we proposed novel evaluation metrics for model explainers that measure the extent that functional specialization and integration are captured by the potential biomarkers.
- We demonstrated that models trained on the salient features identified by the best predictor-explainer combination - as determined by RE-CONFIRM - outperformed other combinations in terms of generalisation performance to out-of-sample datasets, showing that the discovered biomarkers are indeed more robust.

## II. METHODS

GNNs can be applied to FC datasets via two approaches: brain graphs and population graphs. In this study, we focus on evaluation metrics for explainers applied to neural network architecture that incorporate GNN via the brain graph approach. We note that many of the explainers used apply to general deep learning architectures too. Given the functional connectome as $G = (\Omega, \boldsymbol{W})$ where $\Omega$ denotes the set of functionally relevant regions of interests (ROI) or nodes of the network and $\boldsymbol{W}$ denotes the FC matrix derived from the fMRI scan, we further define $f$ as a trained GNN model and $e$ as the explainer that produces a set of saliency scores for $G$.

Our proposed RE-CONFIRM framework uses the input graph, GNN, model predictions, and saliency scores to produce

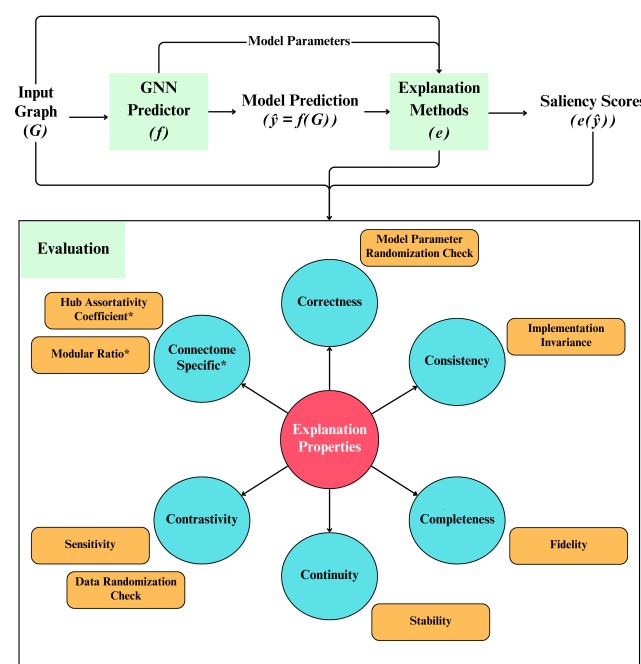

Fig. 1. Our proposed framework, RE-CONFIRM, computes 8 metrics to evaluate the robustness of the saliency scores produced by explainers. Rectangles marked with an asterisk(*) represent metrics proposed in this study that are specific to connectomes.

a range of metrics that evaluates the correctness, consistency, completeness, continuity, contrastivity as well as connectome-specific properties of the saliency scores. Figure 1 illustrates the role RE-CONFIRM plays in typical biomarker discovery pipelines. It also provides an overview of existing and novel evaluation methods used in this study.

### A. Distance measures

Many metrics used in the RE-CONFIRM framework involve a comparison of differences between two sets of values (e.g. making comparisons between healthy controls and patients or studying the differences before and after a change is made to the data or model). In such scenarios, we measured the changes in explanations quantitatively using Hellinger distance. In the case where $P$ and $Q$ are two discrete probability distributions, Hellinger distance can be computed by the following equation:

$$H(P,Q) = \frac{1}{\sqrt{2}}\sqrt{\sum_i \left(\sqrt{P(i)} - \sqrt{Q(i)}\right)^2}$$

where $i$ represents a data point sampled from the distribution. The probability distributions were obtained from the saliency scores.

### B. Evaluation metrics

Building on top of the Co-12 explanation quality properties proposed by Nauta et al. [11], we propose the following set of evaluation metrics for FC biomarker discovery.

*1) Model Parameter Randomization Check (MPRC):* Model weights are perturbed by introducing randomized changes or reinitializing weights (by calling an initialization function) in anticipation of changes in the model's explanation. If the explanations remain unchanged despite these perturbations, it suggests that the explanations do not align with the underlying reasoning of the model. This "sanity check" assesses the faithfulness and sensitivity of the explanation to the predictive model. Higher MPRC is desired (close to 1.0, Range [0,1]).

*2) Data Randomization Check (DRC):* This model-agnostic approach evaluates whether explanations reflect the learned input-output mapping. The data labels for the training samples are altered randomly and a new model is trained on this randomized dataset. When a model is trained on a dataset with randomized labels, it memorizes these labels rather than learning meaningful patterns. Consequently, its performance on new, unseen data is no better than random guessing. Hence, explanations from a model trained on randomized data should differ significantly from those derived from a model trained on the original dataset. Higher DRC is desired (close to 1.0, Range [0,1]).

*3) Fidelity:* This metric quantifies the faithfulness of explanations to the model's predictions. It measures the difference in predicted probabilities by comparing the original predictions with new predictions after removing the top $k$ features (most important, Fidelity+) or the bottom $k$ features (least important, Fidelity-). A higher Fidelity+ (close to 1.0, Range [0,1]) and a lower Fidelity- (close to 0.0, Range [0,1]) is desirable.

$$F^+ = \frac{1}{N}\sum_i (f(G_i) - f(G_i^+)),$$

$$F^- = \frac{1}{N}\sum_i (f(G_i) - f(G_i^-)),$$

where $f(\cdot)$ is the model under study, $G_i$ represents the FC matrix of subject $i$ from a dataset of $N$ subjects whereas $G_i^+$ represents the subgraph for subject $i$ with the highest saliency node features removed and $G_i^-$ represents the subgraph with the lowest saliency node features removed.

*4) Stability:* It measures a model's ability to generate the same explanations before and after perturbations (e.g. adding Gaussian noise) to the model input. Before evaluating the similarity of explanations, it must be ensured that the output of the model stays the same for a slightly perturbed input. We then measure the similarity of the model explanations for the perturbed data to those from unperturbed data. Higher Stability is desired (close to 1.0, Range [0,1]).

*5) Target Sensitivity (Sens):* This metric is based on the intuition that class-specific features from an explanation should vary between classes. Here, we analyze the difference in explanations between disease subjects and typical controls. Lower Sens is desired (close to 0.0, Range [0,1]).

*6) Implementation Invariance (II):* This approach emphasizes the consistency of model explanations by asserting that two models providing identical outputs for all inputs, regardless of their underlying implementations, should also produce identical explanations. To assess whether the explanation method is invariant to specific implementations of the model, we compare explanations from random initializations of the predictive model using two different seeds. Lower II is desired (close to 0.0, Range [0,1]).

*7) Modular Ratio (MR):* Many diseases are known to affect the modularity of functional brain networks [18], [19]. Studying the distribution of saliency scores of nodes within the same modules could reveal disease-driven changes in functional specialization in the brain. Given $G$, a modularization finds a set $S = \{S_i : v_k \in S_i\}_{i=1,k\in\Omega}^{i=L}$ where each element $S_i$ corresponds to a functional module containing sets of nodes and $L$ denotes the number of modules. Assuming that the explainer generates a saliency mask $\boldsymbol{X} \in \mathbb{R}^{|\Omega|\times C}$ where $C$ is the size of the input vector for each node, MR is computed for node $i$ (belonging to module $m$) by:

$$MR(i) = \frac{\frac{1}{|S_m|}\sum_{j\in S_m} d(X_i, X_j)}{\frac{1}{|\Omega|}\sum_{v\in\Omega} d(X_i, X_v)} \tag{1}$$

where $d(\cdot,\cdot)$ represents a function denoting the Manhattan distance between the input vectors. When averaged across nodes, a mean MR value below 1.0 indicates that node $i$ tends to have saliency scores that are more similar to nodes in the same module (relative to all nodes in $G$). Lower MR is desired (close to 0.0, Range [0,1]). We further define MR-$k$ for the setting where $C$ is constrained to the top-$k$ features.

*8) Hub Assortativity Coefficient (HAC):* Many disorders such as ASD are well-known to affect regions of the brain that serve as hubs (i.e. nodes with much higher degrees than others) [20], [21]. Thus, salient features highlighted by explainers would be expected to contain a sizable portion of these hubs. To better quantify this extent, we propose HAC, a metric based on the assortativity coefficient (AC) proposed in Bazinet et al. [22]. AC is defined as the Pearson correlation between cortical thickness annotations of connected nodes. The hub assortativity coefficient is computed with respect to the edge weights $W_{ij}$ between nodes $i$ and $j$ in the FC matrix $\boldsymbol{W}$.

$$HAC = \frac{1}{2m}\sum_{ij} W_{ij} X_i Y_j$$

where $2m$ represents the sum of edge weights. $X_i$ and $Y_j$ represent the saliency score and hub score (z-score within a univariate normal distribution, focusing on nodes with significantly high intra-modular connections [15]) of nodes $i$ and $j$, respectively. A higher HAC (close to 1.0, Range [0,1]) suggests that the saliency scores effectively capture properties related to brain hubs, indicating that the model accurately identifies and relies on central, highly connected regions within the brain network for disease classification. Like MR, HAC-$k$ is defined as a version of HAC constrained to top-$k$ features.

*C. Selecting the best predictor-explainer combinations*

Out of the 8 metrics above, several of them (e.g. MPRC, DRC, Stability, Sens, II) are generic while metrics such as Fidelity+, MR-$k$ and HAC-$k$ are specific to top-$k$ features (thus,

well-suited for biomarker discovery applications). Thus, to choose the best combinations, we propose a 2-step approach.

1) Eliminating poor combinations via sanity checks: combinations with anomalously poor performance on any of these generic metrics (with statistically significant differences from others) should be removed.

2) Choosing the optimal combination via metrics based on top-k features: The choice of metric depends on the research question and the pre-existing knowledge on hand. For instance, if little understanding is available about the disorder, Fidelity+ is used to rank the combinations. On the other hand, if modularity is known to be associated with the disorder, MR-$k$ is used for ranking.

Subsequently, robustness of the selected combination can be confirmed by showing that using the top-$k$ features (from the best combination) to train a separate model leads to better generalization to unseen datasets, as compared to using another set of top-$k$ features by other poorly performing combinations.

## III. RESULTS

### A. Dataset

The ABIDE I dataset contains 387 resting-state fMRI scans from individuals diagnosed with ASD and 436 typical controls, collected from 20 sites. Data from the ADHD-200 dataset was used to further validate our findings. It contains rs-fMRI scans from 279 subjects diagnosed with ADHD and 488 age-matched typical controls. These were collected from 4 sites (NI, NYU, OHSU, and PKU). Preprocessed rs-fMRI data were downloaded from the Preprocessed Connectome Project. Data from the C-PAC pipeline and Athena pipeline were selected for ABIDE and ADHD-200, respectively. Craddock atlas [23] was used to identify 200 ROIs. We computed the mean time series of all voxels within a sphere of radius 2.5 mm around each ROI. Functional connectivity (FC) matrices were computed by determining the Pearson correlation between the mean activation time series for each ROI pair.

### B. Experiment Setup

To assess the effectiveness of our framework, we implemented various disease-specific models for both static and dynamic FC. Static FC represents average functional organization of the brain measured over neuroimaging recordings lasting several minutes. In contrast, dynamic FC captures brain's ability to transition through different functional connectivity configurations on much shorter timescales, typically on the order of seconds. These models include BrainGNN [10], STGCN [24], STAGIN [25], as well as GCN [26] and GAT [27], with the latter two implemented using the BrainGB framework [28]. We incorporated two classes of explainability methods representing post-hoc explainability (GNNExplainer [29]) and intrinsically interpretable methods (Attention and Node Pooling scores). Gradient descent was done using the Adam optimizer with a learning rate of 0.001. Both datasets were split into train-validation-test in the ratio 6:2:2. We used 5-fold cross-validation across all models for both ABIDE and ADHD-200 data. For STGCN and STAGIN, we set the sliding

window size to 60s and stride to 1s. Default model parameters were used to train all the models. In our experiments, we utilized a batch size of 4, which required approximately 2900 MiB of GPU memory. Experiments were carried out across 5 seeds. GCN and GAT had the highest classification accuracy for ABIDE and ADHD respectively. Detailed model performances are presented in Table V.

In our implementation of the metrics, scores were computed for each fold. Mean and standard deviation across all folds are then reported. Hellinger distance was used to compute MPRC, DRC, Stability, Sens, and II. For Fidelity, we focused on the top 20 features. To evaluate stability, we introduced a small amount of Gaussian noise (two standard deviations) to the input node features and perturbed graph data by rewiring the edges between second-degree nodes [13]. We then measured the similarity between explanations using cosine similarity. For MR, the Craddock atlas, known for its probabilistic mapping of brain regions, lacks well-defined labels for ROIs. To address this, we generated new labels by leveraging spatial proximity to established references. Specifically, we calculated the center of mass for each ROI in the Craddock atlas and used these coordinates to identify the nearest corresponding ROIs within the Power atlas [30]. Modules from these new labels were used to calculate MR for all ROIs, as well as only the top 20 significant ROIs (MR-20). For HAC and HAC-20, hub scores were computed via the sum of ambivert degree and participation coefficient [15]. For dynamic FC models, saliency scores were computed for each sliding window segment. These scores were then averaged before the evaluation metrics were applied.

### C. Comparison of Metrics

Saliency scores can be influenced by a multitude of factors: (i) the base model; (ii) the explainer; and (iii) the dataset. To disentangle these factors, we study the variation of the eight RE-CONFIRM metrics across each factor.

*1) Performance across models:* Table I and II summarises the metrics across models for both ABIDE and ADHD-200 (in the whole dataset setting) for GNNExplainer. We found that among dynamic FC models, STAGIN is better than STGCN in all the metrics (including classification accuracy). This suggests that the STAGIN is a more robust predictive model compared to STGCN. Notably, STAGIN also outperforms the three static FC models. Across the three static FC models, no definitive trend emerges, although GCN and GAT exhibit marginally better performance than BrainGNN on certain metrics in both the ABIDE and ADHD-200. They demonstrate greater stability, reduced Fidelity-, lower target sensitivity, and lower II values. However, they suffer from decreased DRC and Fidelity+ scores. Models that incorporated attention mechanisms in their architecture (e.g., STAGIN and GAT) were found to generally perform better than models without attention: they exhibit higher MPRC and HAC scores, along with lower Fidelity- and II values.

*2) Performance across explainers:* Table III and IV summarise the metrics across explainers for both ABIDE and

TABLE I

EVALUATION METRICS FOR GNNEXPLAINER ON ABIDE.
* INDICATES THAT VALUE IN BOLD FOR THE ROW IS SIGNIFICANTLY
GREATER THAN GIVEN VALUE (STUDENT'S T-TEST, P-VALUE ¡ 0.05).

| Metrics | BrainGNN | GCN | GAT | STGCN | STAGIN |
|---|---|---|---|---|---|
| MPRC (↑) | 0.47±0.01* | 0.51±0.01* | 0.58±0.01 | 0.40±0.02* | **0.60±0.03** |
| DRC (↑) | 0.46±0.01 | 0.20±0.01* | 0.27±0.03* | 0.17±0.01* | **0.47±0.03** |
| Fidelity+ (↑) | **0.60±0.02** | 0.36±0.02* | 0.53±0.02* | 0.48±0.01* | 0.49±0.01* |
| Stability (↑) | 0.99±0.02 | **1.00±0.00** | **1.00±0.00** | **1.00±0.00** | **1.00±0.00** |
| HAC (↑) | 0.18±0.01* | 0.20±0.00* | 0.33±0.02 | 0.28±0.02* | **0.34±0.02** |
| Fidelity- (↓) | 0.32±0.02* | 0.29±0.01* | **0.22±0.01** | 0.37±0.01* | 0.30±0.01* |
| Sens (↓) | 0.38±0.01* | **0.31±0.06** | 0.38±0.01* | 0.50±0.06* | 0.34±0.03 |
| II (↓) | 0.19±0.02* | **0.11±0.02** | 0.13±0.01* | 0.18±0.02* | 0.18±0.02* |
| MR (↓) | **0.49±0.02** | 0.52±0.02* | 0.50±0.01 | 0.52±0.03* | 0.52±0.01* |
| MR-20 (↓) | 0.49±0.01* | 0.47±0.02* | 0.48±0.01* | 0.49±0.01* | **0.43±0.01** |

TABLE II

EVALUATION METRICS FOR GNNEXPLAINER ON ADHD-200.
* INDICATES THAT VALUE IN BOLD FOR THE ROW IS SIGNIFICANTLY
GREATER THAN GIVEN VALUE (STUDENT'S T-TEST, P-VALUE ¡ 0.05).

| Metrics | BrainGNN | GCN | GAT | STGCN | STAGIN |
|---|---|---|---|---|---|
| MPRC (↑) | 0.49±0.01* | 0.25±0.01* | 0.42±0.01* | 0.25±0.04* | **0.55±0.03** |
| DRC (↑) | **0.51±0.02** | 0.43±0.03* | 0.44±0.01* | 0.37±0.00* | 0.44±0.02* |
| Fidelity+ (↑) | **0.70±0.01** | 0.32±0.01* | 0.49±0.02* | 0.32±0.03* | 0.62±0.02* |
| Stability (↑) | 0.99±0.00 | **1.00±0.00** | **1.00±0.00** | **1.00±0.00** | **1.00±0.00** |
| HAC (↑) | 0.19±0.01* | 0.26±0.03* | 0.23±0.01* | 0.17±0.03* | **0.33±0.02** |
| Fidelity- (↓) | 0.38±0.01* | 0.29±0.01* | **0.10±0.01** | 0.22±0.02* | 0.18±0.01* |
| Sens (↓) | 0.40±0.01* | 0.27±0.02 | 0.35±0.01* | 0.51±0.04* | **0.26±0.03** |
| II (↓) | 0.12±0.03* | 0.26±0.01* | **0.07±0.01** | 0.12±0.04* | 0.10±0.01* |
| MR (↓) | 0.50±0.01 | **0.49±0.02** | 0.50±0.01 | 0.52±0.01* | 0.50±0.01 |
| MR-20 (↓) | 0.50±0.01* | 0.47±0.01 | 0.46±0.00 | 0.49±0.01* | **0.45±0.02** |

TABLE III

EVALUATION METRICS ACROSS EXPLAINERS ON ABIDE.
* INDICATES THAT THE VALUE IN BOLD FOR ROW IS SIGNIFICANTLY
GREATER THAN GIVEN VALUE (STUDENT'S T-TEST, P-VALUE ¡ 0.05).

| Metrics | GNNExplainer | | Attention | |
| | GAT | STAGIN | GAT | STAGIN |
|---|---|---|---|---|
| MPRC (↑) | 0.58±0.01* | 0.60±0.03* | 0.43±0.02* | **0.67±0.01** |
| DRC (↑) | 0.27±0.03* | **0.47±0.03** | 0.23±0.02* | 0.43±0.03 |
| Fidelity+ (↑) | 0.53±0.02* | 0.49±0.01* | 0.50±0.01* | **0.62±0.02** |
| Stability (↑) | **1.00±0.00** | **1.00±0.00** | **1.00±0.00** | **1.00±0.00** |
| HAC (↑) | 0.33±0.02 | **0.34±0.02** | 0.24±0.02* | 0.27±0.01* |
| Fidelity- (↓) | 0.22±0.01 | 0.30±0.01* | **0.19±0.06** | 0.33±0.02* |
| Sens (↓) | 0.38±0.01* | **0.34±0.03** | 0.36±0.02* | 0.37±0.03* |
| II (↓) | **0.13±0.01** | 0.18±0.02* | 0.16±0.02* | 0.20±0.01* |
| MR (↓) | **0.50±0.01** | 0.52±0.01* | 0.51±0.01 | 0.54±0.02* |
| MR-20 (↓) | 0.48±0.01* | 0.43±0.01* | **0.41±0.03** | 0.46±0.01* |

TABLE IV

EVALUATION METRICS ACROSS EXPLAINERS ON ADHD-200.
* INDICATES THAT VALUE IN BOLD FOR ROW IS SIGNIFICANTLY
GREATER THAN GIVEN VALUE (STUDENT'S T-TEST, P-VALUE ¡ 0.05).

| Metrics | GNNExplainer | | Attention | |
| | GAT | STAGIN | GAT | STAGIN |
|---|---|---|---|---|
| MPRC (↑) | 0.42±0.01* | 0.55±0.03* | 0.55±0.03* | **0.64±0.03** |
| DRC (↑) | 0.44±0.01* | 0.44±0.02* | **0.47±0.02** | 0.43±0.03* |
| Fidelity+ (↑) | 0.49±0.02* | **0.62±0.02** | 0.51±0.01* | 0.47±0.01* |
| Stability (↑) | **1.00±0.00** | **1.00±0.00** | **1.00±0.00** | **1.00±0.00** |
| HAC (↑) | 0.23±0.01* | **0.33±0.02** | 0.31±0.00* | 0.29±0.03* |
| Fidelity- (↓) | **0.10±0.01** | 0.18±0.01* | 0.29±0.02* | 0.19±0.02* |
| Sens (↓) | 0.35±0.01* | **0.26±0.03** | 0.27±0.01 | 0.29±0.02 |
| II (↓) | **0.07±0.01** | 0.10±0.01* | 0.10±0.01* | 0.09±0.01* |
| MR (↓) | 0.50±0.01* | 0.50±0.01* | **0.47±0.02** | 0.49±0.01* |
| MR-20 (↓) | 0.46±0.00* | 0.45±0.02 | **0.43±0.02** | 0.45±0.03 |

ADHD-200. GNNExplainer generally excels across most metrics in both ABIDE and ADHD-200. Compared to Attention, GNNExplainer generates explanations that are generally more robust, consistent across different seed initializations (model weights), and better at discriminating class-specific features. Conversely, explanations generated by Attention tend to be more faithful and less sensitive to parameterization changes in the predictive model, particularly with ADHD-200.

For BrainGNN, we compared evaluation metrics between the model's learned node pooling weights and the explanations produced by GNNExplainer on both the ABIDE and ADHD-200 (Table VII). The explanations generated by GNNExplainer outperformed those derived from the model's intrinsic node pooling weights across most metrics except DRC, Sens and II. However, differences between the three metrics were small.

*3) Performance across datasets:* Based on Table III and IV, there is a lack of consistent trends across all metrics and models when comparing ABIDE and ADHD-200 datasets. Both datasets exhibit stability under minor perturbations. However, ADHD-200 generally shows lower II compared to ABIDE. In ABIDE, hubs appear more prominently among the top-ranked features (based on saliency scores) than in ADHD-200. Explanations derived from ADHD-200 seem to reflect the dataset's intrinsic properties rather than being influenced by varying model architecture implementations or artifacts. From Table I and II, ADHD-200 appears to have slightly lower MR compared to ABIDE. MR scores decreased further for ABIDE than ADHD-200 when focusing exclusively on the top 20 significant ROIs, compared to considering all ROIs. This trend suggests that nodes with low MR scores exhibit saliency scores similar to their neighboring nodes. Therefore, the salient features in the ABIDE dataset exhibit a more pronounced modular relationship compared to those in ADHD-200 (i.e. nodes in the same module tend to have similarly high scores).

### D. Biomarker analysis

We focus the discussion of potential biomarkers on models with the best disorder classification performance, i.e. STAGIN (ABIDE, dynamic), GCN (ABIDE, static), STAGIN (ADHD, dynamic), GCN / GAT (ADHD, static). 25 of the most salient connections (representing the top 0.1% of features) are presented in each chord diagram in Figure 2 (ABIDE) and Figure 3 (ADHD).

For ASD, biomarker analysis on STAGIN via GNNExplainer (Figure 2(a)) revealed salient connections between the somatomotor network (SM) and default mode network (DMN) as well as between SM and the visual network. Intra-modular connections within the visual network were also found to be salient. Similar salient connections were found when attention was used as the explainer (Figure 2(b)), but the specific ROIs largely differ (i.e. different ROIs from the same modules were involved). Each explainer also has several unique inter-module connections that are mostly isolated (i.e. only one salient connection between those modules).

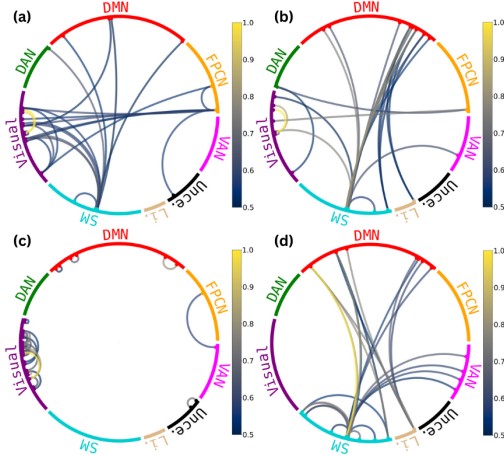

Fig. 2. Chord diagrams representing salient FC connections of ABIDE. In each chord diagram, only the top 25 connections were visualized to reduce cluttering. (a) STAGIN, GNNExplainer, (b) STAGIN, Attention, (c) GCN, (d) GAT, Attention.

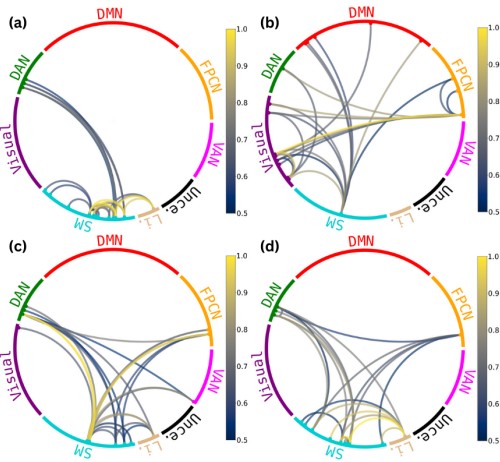

Fig. 3. Chord diagrams representing salient FC connections of ADHD. In each chord diagram, only the top 25 connections were visualized to reduce cluttering. (a) STAGIN, GNNExplainer, (b) STAGIN, Attention, (c) GCN, (d) GAT, Attention.

For static FC, the chord diagram of GCN (Figure 2(c)) was dominated by intra-modular connections within the visual network and DMN. There was an apparent paucity of intra-modular salient connections. Notably, this was also observed when GNNExplainer was applied to GAT, while the use of attention (Figure 2(d)) resulted in a chord diagram with more inter-modular connections. Prominent inter-modular connections include SM-DMN (also in STAGIN) as well as unique connections such as SM - ventral attention network (VAN) and SM - limbic network. However, considering the low performance of GAT and the absence of these connections in other models, these unique inter-modular connections could represent spurious biomarkers.

For ADHD, biomarker analysis on STAGIN via GNNExplainer (Figure 3(a)) revealed salient connections that are concentrated in 3 areas: between SM and dorsal attention network (DAN), between SM and limbic network as well as intra-modular connections within SM. On the other hand, the use of the attention explainer on STAGIN (Figure 3(b)) resulted in widespread inter-modular connections that are salient, differing greatly from GNNExplainer even though the same base model was used (STAGIN).

For static FC (Figure 3(c)), the 3 salient connections found in STAGIN (GNNExplainer) were still present, but another prominent set of connections between SM and frontoparietal control network (FPCN) was found. Attention scores in GAT (Figure 3(d)) shared similar findings (model performance of GAT was very similar to GCN in the case of ADHD, unlike in ABIDE), but with a different weighting: inter-modular connections between SM and Limbic network were emphasized, instead of SM-DAN and SM-FPCN as seen in GCN.

Finally, when comparing salient static FC and dynamic FC features, it was observed that the top connections in ADHD subjects are relatively similar to ABIDE. SM-DAN and SM-Limbic networks were present in both cases for ADHD, while the salient features for ABIDE are markedly different

with static FC dominated by intra-modular connections while dynamic FC highlighted numerous inter-modular connections.

### E. Evaluating the robustness of discovered biomarkers

TABLE V
CLASSIFICATION PERFORMANCE BY THE SVM MODEL ACROSS ABIDE II SITES (BNI, N=58 ; EMC, N=54 ; GU, N=106 ; IP, N=56). * INDICATES THAT VALUE IN BOLD FOR THE ROW IS SIGNIFICANTLY GREATER THAN GIVEN VALUE (STUDENT'S T-TEST, P-VALUE ¡ 0.05) .

| Sites | Attn | GNNExp |
|-------|------|--------|
| BNI | **0.66±0.01** | 0.63±0.01* |
| EMC | **0.65±0.02** | 0.64±0.01 |
| GU | **0.63±0.02** | 0.58±0.02* |
| IP | **0.64±0.01** | 0.62±0.02 |

To demonstrate robustness, we used unseen portions of the ABIDE dataset (i.e. ABIDE-II). We chose Table III for further analysis as it contains variations of both predictors and explainers. First, application of sanity checks based on generic metrics led to the elimination of GAT-related combinations due to anomalously low DRC. Then, we note that STAGIN + Attn has highest Fidelity+ (0.62) while STAGIN + GNNExplainer has the lowest Fidelity+ (0.49). Subsequently, we trained a support vector machine (SVM) with a sigmoid kernel on the ABIDE-I dataset using the top 20 features selected from both combinations and evaluated them on 4 sites from ABIDE-II. The SVMs were tuned on ABIDE-I via 5-fold cross validation with the following ranges: c = $\{0.1, 0.01, 0.001\}$, coef0 = $\{0.1, 0.01, 0.001\}$, gamma = $\{10^{-1}, 10^{-3}, 10^{-5}\}$.

It is clear from Table V that the features from STAGIN + Attn consistently leads to higher out-of-sample accuracies across multiple ABIDE-II sites. This shows that RE-CONFIRM can choose the best predictor-explainer combination that leads to robust biomarkers.

## IV. DISCUSSION

Intuitively, achieving a good model performance is a key prerequisite for discovering robust biomarkers. However, our results revealed that biomarkers produced by the most performant ML techniques still vary widely across explainers. This warrants a more careful interpretation of the 'top-$k$ features' with the highest saliency scores that existing work often report.

In view of this, the current practice of referring to existing literature (i.e. highlighting previous studies that found the same salient features) is insufficient to justify the relevance of these potential biomarkers. For example, SM-DAN was found to be impaired in ADHD subjects in a previous study [31] and this was highlighted by GNNExplainer for STAGIN. However, we note that such cross-referencing could also be easily and superficially performed to justify findings such as SM-DMN being implicated in ADHD [32] (identified by attention explainer, for STAGIN) which was not reproduced by GNNExplainer as one of the most salient features. As such, we would advocate for objective and quantitative metrics that could supplement such qualitative justifications of potential biomarkers. The metrics proposed in the RE-CONFIRM framework could serve as a first step towards this direction.

The eight metrics in RE-CONFIRM revealed that models incorporating attention tend to have higher MPRC, HAC, lower Fidelity- and lower II. The use of GNNExplainer, instead of attention explainer or pooling, could further improve the robustness of these salient features. These metrics provide additional insights, on top of classification accuracy, to determine which combinations of predictors and explainers should be used. Also, in the event that different explainers (applied to the same model) produce contradictory salient features, RE-CONFIRM also makes it possible to suggest which set of potential biomarkers is more robust. For instance, the higher Fidelity+ score achieved by GNNExplainer (0.62 vs 0.47) suggests that the explanations produced by GNNExplainer are more reliable than attention.

One potential source of variations in results could be the choice of $k$ (e.g. Fidelity). Thus, we conducted additional experiments by evaluating the Fidelity metric across different $k$ salient features using STAGIN with GNNExplainer on the ABIDE dataset. From Table VIII, we observed that higher $k$ values lead to greater Fidelity+ and lower Fidelity-. This highlights how the number of salient features affects the evaluation of the explanation. Additionally, we have also explored limiting MR to the top 20 features, revealing how these features have lower MR scores (than the scenario where all features were used). MR values could also depend on the choice of modules. Thus, we conducted additional experiments and from Table IX, we also noticed that varying levels of modularization impact the MR. We applied the ICSC algorithm [33], which provides a finer granularity in module generation for brain functional networks. Using modules produced by the ICSC algorithm leads to lower MR scores (than using modules assigned in the Power atlas) in both static and dynamic FC settings, indicating that nodes within finer group-level modules exhibit more consistent saliency scores compared to those from higher-level modules.

Future research could explore ways to improve the metrics used in RE-CONFIRM. For example, recent studies have highlighted concerns regarding the underlying assumptions in the metric MPRC, particularly regarding the order of layer randomization and the choice of pairwise similarity measures [34]. Solutions to these issues include a smoothed form of MPRC [35], which could be used in place of the original MPRC. Future studies could also propose new metrics. In this study, our assessment of the robustness of model explanations centered on metrics derived from static FC. However, metrics could be specifically tailored for dynamic FC. Instead of merely averaging saliency scores across sliding window segments, these metrics should aim to capture the nuanced temporal dynamics inherent in dynamic FC, thereby providing deeper insights into brain network activity.

In conclusion, our study revealed that salient features identified by ML models are not necessarily robust. Future disease classification studies that generate potential biomarkers could use RE-CONFIRM to reaffirm the robustness of the salient features identified by their models.

## ACKNOWLEDGMENT

This research is supported by AcRF Tier-2 grant MOE T2EP20121-0003 of Ministry of Education, Singapore.

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

## APPENDIX

### TABLE VI
COMPARISON OF MODEL ACCURACIES (%) ON ABIDE AND ADHD DATASETS (sFC=STATIC FC, dFC=DYNAMIC FC).

| Models | ABIDE | ADHD | Selective Model Parameters | Model size |
|---|---|---|---|---|
| BrainGNN (sFC) | 60.8 ± 5.63 | 57.7 ± 5.41 | Epochs=100, GNN Layers=2 | 93k |
| GCN (sFC) | **73.3 ± 3.79** | 68.7 ± 4.62 | Hidden Dim.=360, GCN Layers=2 | 593k |
| GAT (sFC) | 66.3 ± 3.29 | **68.8 ± 3.72** | Hidden Dim.=8, GAT Layers=2 | 655k |
| STGCN (dFC) | 58.8 ± 6.01 | 58.2 ± 6.23 | Window Size=60, Dropout=0.5 | 208k |
| STAGIN (dFC) | 72.4 ± 3.37 | 66.8 ± 2.80 | Hidden Dim.=128, GIN Layers=4,2 | 1,004k |

### TABLE VII
EVALUATION METRICS FOR BRAINGNN USING ITS POOLING WEIGHTS AND GNNEXPLAINER ON ABIDE DATASET

| Metrics | ABIDE | | ADHD-200 | |
|---|---|---|---|---|
| | GNNExplainer | Pooling Weights | GNNExplainer | Pooling Weights |
| MPRC (↑) | 0.47±0.01 | 0.44±0.04 | **0.49±0.01** | 0.48±0.03 |
| DRC (↑) | 0.46±0.01 | 0.47±0.03 | 0.51±0.02 | **0.53±0.02** |
| Fidelity+ (↑) | 0.60±0.02 | 0.53±0.01 | **0.70±0.01** | 0.62±0.01 |
| Stability (↑) | **0.99±0.02** | 0.93±0.03 | **0.99±0.00** | 0.94±0.02 |
| HAC (↑) | 0.18±0.01 | 0.15±0.00 | **0.19±0.01** | 0.17±0.04 |
| Fidelity- (↓) | **0.32±0.02** | 0.34±0.02 | 0.38±0.01 | 0.37±0.02 |
| Sens (↓) | 0.38±0.01 | **0.36±0.01** | 0.40±0.01 | 0.41±0.03 |
| II (↓) | 0.19±0.02 | 0.16±0.01 | 0.12±0.03 | **0.10±0.01** |
| MR (↓) | **0.49±0.02** | 0.51±0.02 | 0.50±0.01 | 0.51±0.03 |
| MR-20 (↓) | **0.49±0.01** | 0.50±0.02 | 0.50±0.01 | 0.50±0.01 |

### TABLE VIII
EVALUATION OF FIDELITY FOR DIFFERENT $k$ FEATURES WITH STAGIN AND GNNEXPLAINER ON ABIDE DATASET.

| Metrics | k=20 | k=80 | k=140 | k=200 |
|---|---|---|---|---|
| Fidelity+ (↑) | 0.49±0.01 | 0.54±0.01 | 0.58±0.02 | **0.61±0.01** |
| Fidelity- (↓) | 0.30±0.01 | 0.28±0.01 | 0.25±0.01 | **0.24±0.02** |

### TABLE IX
EVALUATION OF MODULAR RATIO ACROSS DIFFERENT MODULARIZATION ON BOTH DATASETS (ATTENTION)

| Metrics | GAT (static FC) | | STAGIN (dynamic FC) | |
|---|---|---|---|---|
| | Power | ICSC | Power | ICSC |
| ADHD-200 | | | | |
| MR (↓) | 0.47±0.02 | **0.44±0.03** | 0.49±0.01 | 0.46±0.02 |
| MR-20 (↓) | 0.43±0.02 | 0.43±0.01 | 0.45±0.03 | **0.42±0.02** |
| ABIDE | | | | |
| MR (↓) | 0.51±0.01 | **0.46±0.02** | 0.54±0.02 | 0.47±0.03 |
| MR-20 (↓) | 0.41±0.03 | 0.39±0.01 | 0.46±0.01 | **0.38±0.02** |