# OpenReview forum: "Robustness of explainable AI algorithms for disease biomarker discovery from functional connectivity datasets"
_IEEE.org/EMBS/BHI/2024/Conference — IEEE BHI'24_

### Official Review · Reviewer_qd18 · 2024-08-12
**Robustness metrics for machine learning graph models applied to fMRI for disease biomarker discovery.**

**Overall Rating:** 7
**Confidence:** 3

**Other Quality Metrics:**

(a) Clarity of writing =fair

 (b) Clinical Significance =fair

(c) Methodological Novelty=poor

(d) Experiments and Results= good

**Questions For The Authors:**

1- In section III-B, please mention how much the test train split. Also, in table V, indicate the # of parameters to be trained for each model. Were there enough data samples to train the deep models? After processing, was the dataset balanced? If not, accuracy is not a good metric.


2- As mentioned in “For dynamic FC models, saliency scores were computed for each sliding window segment. These scores were then averaged before the evaluation metrics were applied.” Is there a tendency that the averaging impacts important saliency scores with large variations? How was the sliding window chosen? Can a different sliding window yield a different score? How do saliency scores vary over different segments?

Minor:
Introduction: “intrinsically interpretable models such as”  “interpret instead of interpretable

Please clarify the difference between dynamic and static FC.

**Strengths:**

1. The paper flows smoothly, and the ideas are presented in a consistent and coherent manner.

2. The work builds on previous studies to quantify the robustness of graph machine learning models on fMRI datasets.

3. The evaluation is conducted systematically, investigating the impact of the model, explainer, and dataset on saliency scores in an organized way.

**Summary Of The Paper:**

The paper derives robustness metrics for disease biomarker discovery with functional neuroimaging focusing on brain functional connectomes. The proposed metrics are evaluated on two datasets, multiple graph machine learning models and 2 explainers. The study finds that attention-based graph neural networks yield more robust results, particularly when used with GNNExplainer.

**Weaknesses:**

1-  Most of the metrics proposed in Section II-B are similar to those in Nauta et al. [11], except for MR and HAC. The justification for MR and HAC is based on hypotheses (e.g., "Many diseases are known to affect the modularity of functional brain networks" and "Salient features highlighted by explainers are expected to contain a sizable portion of these hubs."). These hypotheses should be supported by relevant literature. Additionally, further justification is needed to demonstrate why higher HAC and lower MR scores imply a more robust and reliable model or explainer and why biomarkers derived from the best model/explainer combination would be more robust.

2- In Section II-A, it is unclear how the probability distributions are derived from the data. Are these distributions already known, or are they inferred? Moreover, there is concern about how this metric would perform if the dataset or data points are imbalanced or biased toward a certain group.

3-The proposed HAC metric requires more explanation and motivation to justify its effectiveness for biomarker discovery. Firstly, how does HAC differ from the assortativity coefficient? It is recommended to define the assortativity coefficient first and then express HAC in terms of it. Secondly, what is the scale for values in the saliency map and hub scores, and does it ensure that HAC takes values between 0 and 1? Finally, a reference should be cited for the hypothesis: "Many disorders such as ASD are well-known to affect regions of the brain that serve as hubs (i.e., nodes with much higher degrees than others). Thus, salient features highlighted by explainers would be expected to contain a sizable portion of these hubs." Additionally, is it possible for a saliency map to be accurate for a certain disease but not correlate with the hub map (resulting in a low HAC)?

---

### Official Review · Reviewer_KrqR · 2024-08-14
**A nice attempt at evaluating explainers for functional connectivity research domain**

**Overall Rating:** 7
**Confidence:** 5

**Other Quality Metrics:**

Clarity of writing - Great
Clinical Significance - Poor
Methodological Novelty - Great
Experiments and Results - Great

**Questions For The Authors:**

1. While MR and HAC are described well in methods, the real-world implication of either metric isn’t described well. Experiments including Table IX are good, but what does different MR and HAC mean when selecting a specific graph learning algorithm and a supported XAI method?
2. Is HAC and MR only applicable in FC datasets?
3. How can clinicians or biologists benefit from RE-CONFIRM?

**Strengths:**

1.	I really like the experimental validation and associated descriptions on both datasets.
2.	Descriptions of the novel MR and HAC is good.
3.	The paper is well structured, figures and tables are well described.

**Summary Of The Paper:**

The paper titled “Robustness of explainable AI algorithms for disease biomarker discovery from functional connectivity datasets” presents an interesting take on the evaluation of explainable AI (XAI) features in the context of disease biomarker discovery specifically from the functional connectivity (FC) datasets. The authors propose a framework called RE-CONFIRM that integrates existing and two novel evaluation metrics, Modular Ratio and Hub Assortativity Coefficient, tailored to assess the robustness of salient features identified by XAI methods in the context of brain connectomes. The authors apply this framework to evaluate different Graph Neural Network (GNN) models and XAI methods such as GNNExplainer and attention scores in attention-based GNNs. Evaluation on ABIDE and ADHD-200 datasets reveal unique characteristics of both attention-based and GNNExplainer-based XAI methods.

**Weaknesses:**

1.	While the methodology is described around FC datasets, all metrics other than the proposed MR and HAC are applicable to all sort of graph learning problems and architectures and are described in general graph-learning literature. I believe a work like this will benefit a lot from publications such as: Agarwal, C., Queen, O., Lakkaraju, H. et al. Evaluating explainability for graph neural networks. Sci Data 10, 144 (2023). https://doi.org/10.1038/s41597-023-01974-x.
2.	Deciding an algorithm or XAI method based on RE-CONFIRM currently looks like is convoluted due to the varied experiments and results to be considered. If any recommendation or scoring can be made based on the study, that would have been amazing.
3.	The computational aspect of RE-CONFIRM cannot be ignored. On a large dataset with hundreds of thousands of nodes, GCN, GAT, or any other graph learning model will struggle to train without GPUs with large enough memory. Oftentimes, the large graph needs to be split into smaller subgraphs and processed per batch. No information about the computational complexity, data complexity, etc. are mentioned, which makes it harder to recommend.

---

### Official Review · Reviewer_sGdz · 2024-08-17
**A good start**

**Overall Rating:** 6
**Confidence:** 3

**Other Quality Metrics:**

Clarity of writing: Fair
Clinical Significance: Good
Methodological Novelty: Fair
Experiments and Results: Good

**Questions For The Authors:**

How do you verify the efficacy of these evaluation metrics? For example, the paper mentions that STAGIN is better than STGCN because it has higher values for MPRC and DRC (among other metrics) but how do the methods confirm these are the best ways to evaluate the models?

**Strengths:**

The RE-CONFIRM framework is well contextualized and the purpose of developing such a method is clear because the paper addresses how related explainers require better metrics to quantify the robustness of salient features. Most of the evaluation metrics are clear and it is helpful to have the ranges of values desired for each metric, i.e. “Higher DRC is desired (close to 1.0, Range [0,1]). The paper compares many disease-specific models to test the effectiveness of their framework.

**Summary Of The Paper:**

This paper identifies evaluation metrics relevant to functional connectivity and proposes evaluation metrics to measure the robustness of saliency scores from existing explainers in the context of discovering functional connectivity (FC) biomarkers. They focus on datasets of patients with ASD or ADHD to find that salient features in both disorders are highly involved in functional specialization. The paper proposes a RE-CONFIRM framework to set the foundations for the quantitative evaluation of salient features.

**Weaknesses:**

It would be helpful if the performance of the evaluation metrics were discussed in a more clear way - the current discussion is confusing to follow and requires the reader to guess why the results are important. Based on the section about performance across datasets, it seems like these evaluation metrics are not supplying the paper with clear trends and the paper notes that the explanations from ADHD-200 did not show properties of the varying model architecture implementations. The paper might benefit from exploring a different dataset that provides more justification for these evaluation metrics.

---

### Decision · Program_Chairs · 2024-09-23

Accept